# *FUCA1*: An Underexplored p53 Target Gene Linking Glycosylation and Cancer Progression

**DOI:** 10.3390/cancers16152753

**Published:** 2024-08-02

**Authors:** Die Hu, Naoya Kobayashi, Rieko Ohki

**Affiliations:** 1Department of Pharmacology and Toxicology, University of Toronto, Toronto, ON M5S 1A8, Canada; vhu@oicr.on.ca; 2Laboratory of Fundamental Oncology, National Cancer Center Research Institute, Tsukiji 5-1-1, Chuo-ku, Tokyo 104-0045, Japan; nakobay3@ncc.go.jp; 3Department of NCC Cancer Science, Graduate School of Medical and Dental Science, Tokyo Medical and Dental University (TMDU), 1-5-45 Yushima, Bunkyo-ku, Tokyo 113-8510, Japan

**Keywords:** p53, FUCA1, glycosylation, fucosylation, tumor microenvironment, cell adhesion, tumor trans-endothelial migration, cell signaling pathways

## Abstract

**Simple Summary:**

Cancer is a difficult-to-cure disease with high worldwide incidence and mortality. Among the many changes observed in cancer cells and patient samples is altered glycosylation, a commonly observed modification of biomolecules such as proteins. These glycan structures can dictate protein function, and dysregulation of glycosylation can contribute to tumor migration and metastasis. Thus, manipulation of glycosylation states may be a novel approach to cancer treatment. One target of the well-known tumor suppressor p53 is *FUCA1*, encoding alpha-L-fucosidase, which plays a role in glycosylation, although the exact mechanism linking FUCA1 to cancer is unclear. Investigation into these glycosylation processes and the mechanisms linking the p53-FUCA1 axis to cancer development may provide new insights into this disease and suggest new drug targets for cancer therapies.

**Abstract:**

Cancer is a difficult-to-cure disease with high worldwide incidence and mortality, in large part due to drug resistance and disease relapse. Glycosylation, which is a common modification of cellular biomolecules, was discovered decades ago and has been of interest in cancer research due to its ability to influence cellular function and to promote carcinogenesis. A variety of glycosylation types and structures regulate the function of biomolecules and are potential targets for investigating and treating cancer. The link between glycosylation and carcinogenesis has been more recently revealed by the role of p53 in energy metabolism, including the p53 target gene alpha-L-fucosidase 1 (*FUCA1*), which plays an essential role in fucosylation. In this review, we summarize roles of glycan structures and glycosylation-related enzymes to cancer development. The interplay between glycosylation and tumor microenvironmental factors is also discussed, together with involvement of glycosylation in well-characterized cancer-promoting mechanisms, such as the epidermal growth factor receptor (EGFR), phosphatidylinositol-3-kinase/protein kinase B (PI3K/Akt) and p53-mediated pathways. Glycan structures also modulate cell–matrix interactions, cell–cell adhesion as well as cell migration and settlement, dysfunction of which can contribute to cancer. Thus, further investigation of the mechanistic relationships among glycosylation, related enzymes and cancer progression may provide insights into potential novel cancer treatments.

## 1. Introduction

In recent years, glycosylation has attracted a lot of research interest because of its importance in cancer development and diagnosis, as well as potential cancer therapies. Glycosylation is the enzyme-catalyzed addition of saccharides to biomolecules such as proteins, sphingolipids and free-circulating polypeptides [1], and it plays a major role in modulating cellular functions. Glycosylation can be classified as O- or N-linked, depending on its target [2]. The covalent linking of sugar molecules requires them to be in a positive energetic state, most commonly provided by sugar nucleotides. Examples of such sugar nucleotides include uridine diphosphate N-acetylgalactosamine (UDP-GalNAc) (Figure 1a), uridine diphosphate N-acetylglucosamine (UDP-GlcNAc) (Figure 1a), guanosine diphosphate mannose (GDP-mannose) (Figure 1b) and guanosine diphosphate fucose (GDP-fucose) [2,3,4,5] (Figure 1c).

The main cellular compartments where glycosylation takes place include the endoplasmic reticulum (ER) and the Golgi apparatus, although some glycosylation can occur in the cytosol [1]. These glycosyl modifications can take many diverse forms through the attachment of various types of saccharides, different sugar–sugar and sugar–target linkage types and the generation of complex branching structures, each of which can potentially modulate biological function [2]. The tumor suppressor p53 has been shown to inhibit aerobic glycolysis and protein glycosylation [6]. One target of p53 is *FUCA1*, which encodes alpha-L-fucosidase 1, a hydrolase that plays a role in degrading glycan structures [7]. These observations suggest that characterization of glycan structures and their regulation may provide novel insights into cancer development.

This review describes the various glycosylation processes, the enzymes involved, and their relationship with cancer development. We further discuss the roles of glycosylation in tumorigenesis, progression and metastasis, together with the role of several well-known cancer-associated cell signaling pathways such as epidermal growth factor receptor (EGFR), phosphatidylinositol 3-kinase/protein kinase B (PI3K/Akt) and p53 pathways affected by glycosylation.

## 2. Cancer and Glycosylation

Glycosylation refers to the process of adding carbohydrates (monosaccharides or glycans) to a noncarbohydrate molecule, referred to as an aglycone. Depending on the aglycone type, the resulting glycosylated molecule can be classified as glycoprotein, glycolipid or proteoglycan. The addition of various carbohydrates can modulate the structure and conformation of the target and regulate intermolecular interactions such as cell–cell and cell–ECM (extracellular matrix) interactions. Accordingly, aberrant glycosylation can result in the dysregulation of these interactions and potential pathology [1].

Protein glycosylation occurs post-translationally, mostly in the ER and the Golgi apparatus and sometimes in the cytoplasm or nucleus. Glycoproteins can be further divided into several subtypes, depending on the linkage site. The two most common subtypes are N-linked to asparagine (Asn) or arginine (Arg), and O-linked to serine (Ser) or threonine (Thr) [2].

Depending on the conjugated monosaccharide, O-linked glycans can be of several types, as follows: O-GalNAcylated mucin (UDP-GalNAc), O-GlcNAcylated glycan (UDP-GlcNAc) or O-mannosylated glycan (GDP-mannose) (Figure 2a). High expression of O-glycans has been observed in lung cancer patients and are a consequence of signal transducer and activator of transcription 3 (STAT3) hyper-activation induced by the PI3K/Akt pathway, which has been implicated in oncogenesis [8,9]. The generation of truncated O-glycans, resulting from premature terminal formation, is also associated with cancer progression [1,2]. Sialylation, a process of covalently adding sialic acids—derivatives of neuraminic acids—to elongated glycans, is a common step in generating terminal structures of carbohydrate chains [10]. These antigens are targets for antigen-binding adhesin which mediates cell–cell interactions [2]. Premature O-glycan truncation often leads to heavy sialylation, which causes an increase in net negative charge and consequent steric hinderance of cell–cell interaction, and thereby may lead to metastasis [11,12]. Furthermore, O-GlcNAc transferase (OGT), the enzyme responsible for the initial generation of O-GlcNAcylated glycans (Table 1), has been linked to carcinogenesis. The translocation of nuclear factor kappa B (NF-κB) to the nucleus, and consequent transcriptional activity, is modulated by O-GlcNAcylation [13] (Figure 3). OGT also modulates the function of RNA polymerase II (Pol II), as O-GlcNAcylation can compete with phosphorylation of regulatory sites in the enzyme complex [14,15] (Figure 3). The observed elevation of O-GlcNAcylation in cancer cells may be due to the preference of OGT for unstructured and flexible protein regions, which have a higher abundance in cancer compared to normal cells [16,17,18,19,20].

N-glycosylation serves to localize and direct proteins to secretory pathways, to track proteins within cells, and to regulate cell signaling [21,22,23,24]. A diversity of N-glycan glycoforms (Figure 2b) are involved in cellular and disease processes [25,26]. These processes can be influenced by N-contained side chain exposure, changes in protein conformation and the availability of glycosidase and glycosyltransferases inside the respective subcompartments [24]. N-glycosylation is frequently observed on receptors involved in cellular signaling pathways such as the EGFR pathway, and the biological outcome of these pathways is significantly influenced by the extent of N-glycosylation [23], including an association of N-glycosylation with carcinogenesis.

Glycan extension can also be terminated by special structures such as Lewis antigens (Le^a^, Le^b^, SLe^a^, Le^x^, Le^y^ and SLe^x^) (Figure 2c). Overexpression of Lewis antigens and the resulting loss of cellular adhesion have been observed in cancer samples. For instance, SLe^a^ expression has been used to monitor patient responses to cancer therapy, as detection of SLe^a^ by assay is a clinically approved cancer-associated marker [2]. SLe^x^, on the other hand, has been implicated in mediating cell–cell adhesion as ligands for lectins, and can contribute to tumor cell migration and metastasis when dysregulated [2].

Glycosylated proteins play a role in various cellular processes predominantly as secreted and cell-surface proteins. These aberrant glycan structures can disturb receptor–ligand binding and cell–cell adhesion and thereby skew cellular signaling pathways toward conditions favoring carcinogenesis, invasion and metastasis [27,28,29,30,31,32,33,34,35,36]. As such, cancer-associated glycosylation patterns and expression of glycosylation-related enzymes could serve as potential biomarkers for cancer detection and as potential drug targets for cancer therapeutics.


cancers-16-02753-t001_Table 1Table 1Summary of major glycosylation-related enzymes together with their enzymatic functions, implication in glycosylation process and cancer indication.GeneEnzymeMechanism of ActionGlycosylation ProcessCancer IndicationReference
*OGT*
O-N-acetylglucosamine (O-GlcNAc) transferaseAddition of GlcNAc to serine/threonine (Ser/Thr)O-GlcNAcylation initiationTranscription; cancer epigenetics; cell signaling; carcinogenesis[37]
*OGA*
O-GlcNAcaseRemoval of GlcNAc from Ser/ThrDeGlcNAcylation of O-glycanTranscription; cancer epigenetics; cell signaling; carcinogenesis[37]
*FUT1*
Fucosyltransferase 1α1,2-FucosyltrasferaseLewis antigen Le^b/y^ synthesisCell proliferation; metastasis; invasion; angiogenesis[38,39]
*FUT2*
Fucosyltransferase 2α1,2-FucosyltrasferaseLe^b/y^ antigen synthesisCell migration; invasion; cancer progression[38,39]
*FUT3*
Fucosyltransferase 3α1,3/4-FucosyltrasferaseLe^a/b/x/y^, SLe^a/x^ antigen synthesisCancer progression; poor prognosis; Epithelial-to-mesenchymal transition (EMT)[38,39]
*FUT4*
Fucosyltransferase 4α1,3-FucosyltrasferaseLe^x^, SLe^x^ antigen synthesisCell proliferation; anti-apoptosis; multidrug resistance (MDR)[38,39]
*FUT5*
Fucosyltransferase 5α1,3-FucosyltrasferaseSLe^x^ antigen synthesisCell proliferation; metastasis[38,39]
*FUT6*
Fucosyltransferase 6α1,3-FucosyltrasferaseSLe^x^ antigen synthesisCell proliferation; metastasis; MDR[38,39]
*FUT7*
Fucosyltransferase 7α1,3-FucosyltrasferaseSLe^x^ antigen synthesisCell proliferation; anti-apoptosis[38,39]
*FUT8*
Fucosyltransferase 8α1,6-FucosyltrasferaseCore fucosylation of N-glycansCell proliferation; metastasis; MDR; poor prognosis[38,39]
*FUT9*
Fucosyltransferase 9α1,3-FucosyltrasferaseLe^x^ antigen synthesisCancer stemness; cell proliferation; MDR[38,39,40]
*FUT10*
Fucosyltransferase 10α1,3-FucosyltrasferaseCore fucosylation of N-glycansNot yet observed in human[39,41]
*FUT11*
Fucosyltransferase 11α1,3-FucosyltrasferaseCore fucosylation of N-glycansNot yet observed in human[39,41]
*POFUT1/FUT12*
Protein O-fucosyltransferase 1Transfer fucose to Ser/ThrO-FucosylationHigh expression in cancer samples; invasion; differentiation[39,42]
*POFUT2/FUT13*
Protein O-fucosyltransferase 2Transfer fucose to Ser/ThrO-FucosylationHigh expression in cancer samples; poor prognosis; invasion; differentiation[39,42,43]
*FUCA1*
alpha-L-fucosidase 1Removal of attached fucoseDefucosylationCell proliferation; patient survival[7,39]
*FUCA2*
alpha-L-fucosidase 2Removal of attached fucoseDefucosylationHigh expression in cancer samples; poor prognosis[39]
*ST6GALNACs*
α-N-acetylgalactosaminide (GalNAc) α-2,6-sialyltransferasesα6-Sialylation of O-GalNAcTerminal sialylationSialyl-Thomsen-nouveau (STn) overexpression; cancer prognosis marker; cell proliferation; migration; cell adhesion[2,44]
*ST3GALs*
ST3 β-galactoside α-2,3-sialyltransferasesα3-Sialylation of galactoseTerminal sialylationO-glycan truncation; metastasis; invasion; cell proliferation; cancer prognosis marker[2,44]
*GMDS*
GDP-mannose-4,6-dehydrataseGDP-mannose-4,6-DehydrataseGDP-fucose de novo synthesisHigh expression in cancer samples; anti-apoptosis; EMT[5,45,46]
*GFUS*
Guanosine diphosphate fucose (GDP-L-fucose) synthaseSynthesis of GDP-fucoseGDP-fucose de novo synthesisCell–selectin binding; cell differentiation; cell proliferation; extravasation[47,48]


## 3. Glycosylation and the Tumor Microenvironment

Not only do cancer cells adapt to grow and survive, but they also alter interactions within the local microenvironment to promote tumor maintenance and metastasis. Glycosylation plays an important role in these interactions. The tumor microenvironment (TME) is a complex set of cellular, physical and soluble mediators surrounding the tumor tissue. These include blood and lymph vessels, endothelial cells and immune cells [49,50] as well as acellular components such as cytokines and the ECM [50,51,52]. These cell–stromal interactions can skew the physical and biochemical properties of the TME toward conditions that favor tumor growth and survival.

Dysregulated cell growth subjects cells within tumor tissue to constant stress, including hypoxia (Table 2). The arrangement of blood and lymph vessel surrounding tumors tends to be disorganized, resulting in poor oxygen penetration into the tumor center [53]. As a result, tumor cells may shift from oxidative phosphorylation to aerobic glycolysis [54,55] (Figure 1a). This shift is called the Warburg effect, and it is an adaptation to the limited oxygen availability that serves to maintain the energy required for cell growth [54]. One response induced by hypoxia is the activation of hypoxia-inducible transcription factors (HIFs) and their signaling pathways [56,57,58,59]. These HIFs transcriptionally regulate glycolysis and glycosylation, although exactly how they alter glycosylation is unclear at present [60]. Hypoxia causes a shift toward the biosynthesis of glycans with low α2,6-sialylation and high β1,6-branching, as well as elongation by poly-N-acetyllactosamine (poly-LacNAc) [61] (Figure 2b). This biased generation of glycans leads to weaker cell adhesion, which could potentially facilitate metastasis.

Acidification of the TME is another adaptive strategy to promote the growth of tumor versus normal cells by creating a hostile, low pH extracellular microenvironment [62,63,64,65] (Table 2). TME acidification is attributed to increased aerobic glycolysis and lactate production due to the Warburg effect [55]. Additionally, the disorganization of lymphatic vessels responsible for the degradation of lactic acids also contributes to the lower pH of the TME [63,66]. Moreover, the accumulation of lactic acids contributes to the synthesis of hypoxanthine (a potential oxygen free radical generator) [67], as well as the expression of the cell-surface CD44 antigen [68]. CD44 antigens, which are themselves glycoproteins, function as receptors of proteoglycans (one subtype of glycan) including hyaluronic acid [2,60,69,70] and chondroitin sulfates [71,72], and these interactions are known to be tumor-associated [2]. Thus, TME acidification and the consequent alteration in CD44-mediated cellular interactions provide another example linking glycosylation with cancer progression and tumor metastasis.

In addition, biological molecules involved in cell–cell and cell–matrix adhesion may be altered in the TME, influencing endothelial adhesion, cell mobility and metastasis [60] (Table 3) Altered glycosylation of intercellular adhesion molecule 1 (ICAM1) [73,74], vascular cell adhesion protein 1 (VCAM1) [75] or platelet endothelial cell adhesion molecule (PECAM1) [76] can influence monocyte rolling and cell adhesion (Table 3). Altered glycosylation of glycoproteins, collagen, glycosaminoglycans (GAGs) and proteoglycans (multi-GAG chains) residing in the ECM are also potential mediators of cell–matrix interactions [2].

In summary, a hostile TME favors the survival of tumor cells over normal. Glycosylation plays various essential roles in the establishment of the TME, the adaptation of tumors to this environment and the downstream consequences of these processes. This involves altered communication between cells and the extracellular environment, mediated by glycosylated cellular surface receptors and transmembrane proteins. As such, dysregulation in glycosylation can lead to the loss of cell–cell and cell–matrix contact, which, in turn, can facilitate cancer cell migration from the primary tumor to distant sites [52].
cancers-16-02753-t002_Table 2Table 2Summary of glycosylation alterations that are induced during tumor microenvironment establishment and maintenance.Tumor Microenvironment PropertiesCausationCancer-Promoting FunctionsGlycosylation AlterationHypoxiaPoor oxygen penetration due to suboptimal blood and lymphatic arrangement [53].Shift from oxidative phosphorylation to aerobic glycolysis [54,55];Produces lactate [55,62] to meet energy requirement in tumor cells [54].Alteration in glycosylation by hypoxia-inducible factors (HIFs) [56,57,58,59];Favors biosynthesis of glycans with low α2,6-sialylation, high β1,6-branching and poly-LacNAc structure [61];Promotes galectin-1 production [61].Low pHHigh level of lactate production due to Warburg Effect [55];Low level of lactate degradation due to lack of functional lymphatic vessels [63,66].Hostile to normal cells but favors tumor cell growth [62,63,64,65].CD44 production [68];CD44–proteoglycan binding mediated intercellular interactions [2,69,71].
cancers-16-02753-t003_Table 3Table 3Summary of glycosylated and glycosylation-related molecules that are implicated in tumor microenvironment establishment and maintenance.FunctionGlycosylated & Glycosylation-Related MoleculesCancer IndicationCell adhesionIntercellular adhesion molecule 1 (ICAM1) glycoforms [73,74];SLe^x^ antigens [77,78];Selectins [79];Vascular cell adhesion molecule 1 (VCAM1) glycoforms [75];Platelet endothelial cell adhesion molecule (PECAM1) glycoforms [76].Cell rolling, migration and adhesion [73,74]; cancer progression [27,28,29,30,31,32,33].Trans-Endothelial MigrationGlycosylated epitopes on tumor cells [71,80];SLe^x^ antigen [27,28,29,30,31,32,33];ST3Gals [81].Loss of cell adhesion [60]; cell intravasation, rolling and extravasation [60]; tumor metastasis [82,83,84,85]; cell migration [86].

## 4. p53 and Glycosylation

The gene *TP53* codes for the tumor protein p53—the well characterized “guardian of the genome”—which was originally characterized as a tumor suppressor that activates the transcription of target genes responsible for repairing DNA damage (Figure 4a). However, subsequent studies revealed that p53 function is not limited to the DNA damage response and repair pathways but also extends to the maintenance of energy metabolism [6]. A high rate of intracellular glycolysis and the acidification of the TME due to lactate accumulation induce p53 expression and activation, which functions to protect cells from glycolytic stress [87]. During tumor development, the loss of p53 in cells surrounding the tumors can further acidify the TME and promote carcinogenesis. Moreover, reactive oxygen species (ROS) that are produced during mitochondrial respiration via oxidative phosphorylation are also harmful to cells (Figure 1a) [88,89], and wild-type p53 is upregulated in response to protect cells from high levels of toxic ROS. However, oncogenic mutation of p53 (mt p53) in cancer cells abrogates this protective response against ROS, leading to increased oxidative damage to cells [90].

The inhibition of glycolysis and shift to aerobic metabolism, known as the Warburg effect, are also known to depend on p53 activity [91]. Thus, the loss of p53 function in various cancers further promotes aerobic glycolysis [6], a hallmark of cancer cell metabolism [92]. Increased glucose uptake and lactate production from aerobic glycolysis is considered to be an adaptation of cancer cells to the hypoxia microenvironment created by tumor growth [55,62]. High energy demand, TME acidification and an increase in ROS production can induce the expression and activation of normal p53 [87,88,89]. p53 in turn regulates energy metabolism by modulating the glycolysis pathways [91]. A main effector of this regulation is the *TP53*-induced glycolysis regulatory phosphatase protein (TIGAR), whose expression is induced by activated p53 (Figure 4b) and which functions to reduce levels of fructose-1,6-biphosphate, an intermediate of glycolysis [93]. Limiting the abundance of this substrate results in a reduction in glycolysis, which in turn prevents extensive production of harmful ROS. In contrast, loss of functional p53 is a profound enhancer of aerobic glycolysis [94].

Another link between p53 and cellular energy metabolism is seen in the regulation of NF-κB-dependent transcriptional activity by p53. Mutation of p53 causes enhanced O-GlcNAcylation of IKK, inhibitor of nuclear factor-κB (IκB) kinase, thereby upregulating IKK activity and upregulating NF-κB activity (Figure 4b). IKK is composed of the subunits IKKα and IKKβ, and O-GlcNAcylation of IKKβ occurs at Ser733, which is also the site of inactivating phosphorylation. O-GlcNAc interferes with phosphorylation at this site, enhancing IKK activity and promoting the ubiquitination and removal of IκB. This leads to enhanced NF-κB activation and increased cell survival [95] (Figure 4b). Upregulation of NF-κB also can promote aerobic glycolysis [96] and drive oncogenesis by enhancing expression of solute carrier family 2 member 3 (SLC2A3) [97]. O-GlcNAcylation of IKKβ provides another example of a competition between glycosylation and phosphorylation that can dictate protein function, similar to the example of Pol II regulation mentioned earlier.

Furthermore, p53 itself can be O-GlcNAcylated (Figure 4b). Studies have demonstrated that O-GlcNAcylation of p53 occurs at Ser149, and this modification inhibits ubiquitin-based degradation of p53. Ser149 is proximal to a phosphorylation site at Thr155 and when O-GlcNAcylated can potentially suppress phosphorylation at Thr155 and thereby decrease susceptibility to ubiquitination [98] (Figure 4b). Moreover, Thr155 phosphorylation is necessary for p53 nuclear export mediated by Jun activation domain-binding protein 1 (JAB1) (Figure 4b). Thus, O-GlcNAcylation prevents phosphorylation at Thr155, inhibits p53 nuclear export, inhibits ubiquitination and consequently increases p53 half-life in cells [99,100]. Thus, the p53 protein, which previously has been characterized as a DNA damage response factor, is also implicated in glycosylation. p53 not only modulates glycosylation on other proteins via its effect on glycolysis, but is itself glycosylated in a manner that can affect its function.

Mutated p53 in cancer cells has also been implicated in the aberrant folding of N-glycosylated proteins via its transcriptional regulation of the ectonucleoside triphosphate diphosphohydrolase 5 (*ENTPD5*) gene [101,102]. ENTPD5 protein is a component of the calnexin/calreticulin chaperon system that facilitates N-glycoprotein folding [103,104]. Receptor tyrosine kinases (RTKs), including transforming growth factor beta (TGFβ) receptor and EGFR [105], are highly modified by N-glycans, and their oncogenic mutations are associated with the calnexin/calreticulin chaperon system [103,106]. Moreover, ENTPD5 expression level is highly associated with the presence of mt p53 at both the transcript and protein levels, and knockdown of either *ENTPD5* or mt *p53* attenuates the invasiveness and metastatic potential of cancer cells [102]. These observations indicate that mt p53 can modulate the folding of N-glycoproteins essential for tumorigenesis and cancer progression via ENTPD5 and the calnexin/calreticulin chaperon system in cancer cells (Figure 4b).

Taken together, p53 is involved in the modulation of energy metabolism and generation of protein glycoforms, including modulation of its own activity by glycosylation, illustrating the potential contribution of p53 to cancer development from the energy homeostasis perspective [98].

## 5. Fucosylation in Cancer

Another type of protein glycosylation is fucosylation, which has been extensively studied in human metabolism and has been the subject of much interest due to its apparent involvement in oncogenesis. Fucosylation is classified into terminal (α1-2 or α1-3/4 linked) and core fucosylation (α1-6 linked), depending on the sites of attachment [2]. Unlike GlcNAcylation or GalNAcylation, which are direct linkages to the target, fucosylation decorates previously synthesized glycans with fucose.

### 5.1. Fucose Nucleotide Biosynthesis

GDP-fucose is the sugar donor used for fucosylation. This sugar nucleotide is formed in the cytosol via the following two possible routes: de novo synthesis and salvage pathways. 90% of GDP-fucose comes from the de novo pathway, during which cytosolic GDP-mannose is converted step-wise into GDP-fucose via catalysis by GDP-mannose-4,6-dehydratase (GMDS) [5] and GDP-L-fucose synthase (GFUS or protein FX) [47] (Figure 1c) (Table 1). In contrast, the salvage pathway utilizes free cytosolic fucose to synthesize GDP-fucose via catalysis by fucose kinases and fucose-1-phosphate guanylyltransferase (FPGT) [107] (Figure 1c). GDP-fucose is thereafter transported into the Golgi apparatus and serves as the sugar donor for fucosylation, catalyzed by FUTs [108].

### 5.2. Fucosyltransferases (FUTs) and Fucosidases

The human genome encodes 13 FUTs that are responsible for terminal and core fucosylation with different α-linkages (Table 1). FUTs are mostly located in the following two subcellular compartments: the N-glycan targeting FUTs reside in the Golgi apparatus while O-fucosyltransferases reside in the ER [39]. Two fucosidases (FUCA1 and alpha-L-Fucosidase 2 (FUCA2)) were found to carry out the defucosylation of glycans in humans [39]. The distinct functions of each FUT and fucosidase is listed in Table 1. Fucosylation is enhanced in cancer development [109,110,111,112] and cancer aggressiveness is apparently inhibited when fucosylation is reduced [113] Thus, FUTs and fucosidases may be novel candidate drug targets for cancer therapeutics.

### 5.3. Fucosylation and Cancer

Aberrant fucosylation is frequently observed in cancer samples. Lewis antigen Le^x^ (Figure 2c) is used as a biomarker for glioblastoma multiforme (GBM) due to the correlation between its high expression and the tumorigenic potential of cells [109,110]. FUT8, which attenuates cancer aggressiveness when knocked down [113], is overexpressed at both the mRNA and protein levels in lung and colorectal cancer (CRC) samples and its expression is associated with poor prognosis [111]. In addition, the sialylated Lewis antigens SLe^x^ and SLe^a^ (Figure 2c) are globally elevated in cancer specimens [2] and are implicated in carcinogenesis and tumor metastasis via their ability to enhance cancer cell mobility across the endothelium [114,115]. General upregulation of fucosylation is also observed in lung cancer adenocarcinoma, often accompanied by increased GMDS expression [45]. *FUCA1* mRNA expression is generally lower in cancer samples compared to normal counterparts, and this is associated with poor patient prognosis [7,116]. In contrast, *FUCA2* is proposed to be a cancer prognostic marker due to its high mRNA expression in pan-cancer patients and association with poor survival rates [117]. The correlation between FUCA2 high expression and cancer has also been confirmed at the protein level in lung carcinoma and uterine corpus endometrial carcinoma [117].

Current studies on the contribution of fucosylation to carcinogenesis have revealed its effects on malignant cell proliferation, invasion, metastasis, immune surveillance escape and multidrug resistance (MDR) [39].

### 5.4. Fucosylation in Cancer Cell Proliferation

Fucosylation levels are positively associated with cancer cell proliferation. Maintaining higher fucosylation in cancer cells requires securing higher amounts of its sugar nucleotide donor GDP-fucose. Inhibition of GDP-fucose synthesis leads to decreased fucosylation of receptors implicated in the EGFR signaling pathway and, thus, reduces cancer cell proliferation [118]. In addition, knocking down *GMDS* induces cell cycle arrest and activates apoptosis [45]. FUTs have been shown to promote cell cycle progression [119,120,121,122]. In addition, alterations in *FUT*s’ mRNA expressions result in changes in the levels of Lewis antigen abundance, which interfere with various well-known cancer-associated signaling pathways including the PI3K/Akt [123] and EGFR/mitogen-activated protein kinase (MAPK) pathways [124]. Multiple lines of evidence have also supported the notion that fucosidases are involved in cancer progression and metastasis, as follows: (1) ectopic expression of FUCA1 in samples having low endogenous levels of FUCA1 significantly attenuates the invasiveness of cancer cells; (2) transient *FUCA1* knockdown results in the loss of cell–cell contact; and (3) prolonged *FUCA1* knockdown leads to increased cancer cell proliferation, invasion and migration [125]. Moreover, fucosylation can contribute to the hyperactivation of the EGFR, a well-known driver of carcinogenesis, and this can be reversed by FUCA1 overexpression, which reduces fucosylation [7]. These observations indicate the tumor suppressor potential of FUCA1. In contrast, *FUCA2* expression is commonly elevated in cancer [117,126]. Anti-FUCA2 antibodies inhibit the proliferation of breast cancer cells, and this is reversed by ectopic expression of FUCA2, suggesting that this fucosidase has oncogenic potential [127].

### 5.5. Fucosylation in Cancer Stem Cells

Similarities have been observed between cancer development and embryonic development, and thus important biological pathways for embryogenesis are expected to demonstrate similar significance in cancer progression. Although the exact role of fucosylation in human embryo development has not been fully elucidated, resultant lethal phenotypes in mouse embryos arising from defects in fucosylation hints at the essentiality of this process for proper cell growth and proliferation [128]. This is further supported by fucosidosis, a rare neurodegenerative autosomal recessive disorder, being mediated by *FUCA1* mutation [129].

Cancer stem cells (CSCs), as the postulated origin of most human tumors, have also been proposed to contribute to cancer development. Research in oral squamous cell carcinoma has demonstrated that inhibiting fucosylation negatively affected tumor initiation and CSCs invasion, and fucosylated SLe^x^ antigen is implicated in metastasis associated with CSCs [130]. These observations could be explained by the fucosylation of integrins—cellular surface molecules that function to guide movement and localization of cells through intercellular interactions with glycan-binding proteins (GBPs) presented on other cells. This implication is not limited to cancerous cells. For example, modulating the fucosylation status of integrins is a potential strategy for the delivery of noncancerous multipotent stromal cells to target tissues for tissue repair [131]. Moreover, enhanced fucosylation of cytotoxic T lymphocytes improves their homing to tumor tissue and thus is proposed to be an effective strategy for cancer treatment [132]. These results indicate that fucosylation is also implicated in cancer progression from the aspect of stem cells potentially through its role in guiding cell migration.

### 5.6. Fucosylation in the Epithelial-to-Mesenchymal Transition (EMT)

Fucosylation has been shown to be involved in the EMT. EMT refers to a biological process consisting of the following: (1) the transition of epithelial tumor cells bearing epithelial cellular markers to cells that mimic the mesenchymal cell phenotype and exhibit mesenchymal cellular markers; and (2) the degradation of basement membrane, enhancing the capability of tumor cells to migrate to distant sites [133]. Similar to other types of glycosylation, fucosylation may also affect cell–cell adhesion by modulating the function of EMT factors. Cancer-associated upregulation of *GMDS* and *FUT8* leads to an enhancement in core fucosylation followed by appearance of mesenchymal markers [46], indicating enhanced EMT. Furthermore, transcription of EMT factors is induced by fucosylation of translocation-facilitating proteins, which in turn promote the nuclear translocation of transcription coactivators that skew cells toward the mesenchymal cell phenotype [134]. Additionally, upregulation of fucosylation promotes an elevation in matrix metalloproteinases (MMPs) that degrade the ECM [135], clearing boundaries around tumor cells and facilitating metastasis. EMT can also be induced by signaling pathways such as the TGFβ-dependent cascade, which is regulated by both FUT8 and FUCA1 [136,137]. Expression of *FUCA1* decreases during TGFβ-induced EMT, resulting in the generation of highly fucosylated N-glycans associated with cancer development [136].

### 5.7. Fucosylation and Tumor Cell Trans-Endothelial Migration

Translocation of leukocytes across tissue barriers during the inflammatory response involves glycoprotein-mediated alterations in cell adhesion, and tumors utilize this mechanism to allow trans-endothelial migration and metastasis [71,138,139]. Glycosylated epitopes on tumor cells facilitate cell rolling along the endothelium by binding to GBPs exposed on endothelial cells [71,80] (Figure 5). These interactions between GBPs exposed on the endothelial cellular surface and the glycan structures of circulating biomolecules such as leukocytes and tumor cells are modulated by enzymes such as glucosyltransferases and FUTs [80,140] (Table 3).

Fucosylation reduces cell adhesion, thereby allowing intravasation and extravasation of tumor cells across the endothelium [60]. Specifically, these cellular interactions are modulated by terminal fucosylated glycans binding to lectins. Fucosylation alters the interactions of glycans with lectin, allowing enhanced cell rolling mobility [79]. Similarly, fucosylation allows tumor cells to mimic leukocyte transportation and travel from the primary site to colonize distant parts of the body [27,28,29,30,31,32,33,71,138,139] (Figure 5). This argument is consistent with the observed increase in the abundance of fucosylated SLe^x^ motifs on migrating tumor cells [27,28,29,30,31,32,33].

Various glycosylation-related enzymes contribute to the synthesis of fucosylated glycans. One example of a sialylation-related enzyme implicated in cancer development is ST3 beta-galactoside alpha-2,3-sialyltransferase 6 (ST3GAL6), which is essential for sialylated Lewis antigen synthesis. Knockdown of *ST3GAL6* was observed to attenuate cell trans-endothelial migration ability, while elevation of *ST3GAL6* expression was found to be associated with low overall survival in multiple myeloma patients [81] (Table 1). Reduction in enzymes involved in fucosylation decreases the synthesis of the lectin ligands SLe^a^ and SLe^x^, leading to attenuated binding of tumor cells to lectins [48].

These observations indicate that cancer-associated changes in fucosylation-related enzymes affect the adhesion of cells essential for endothelium integrity, contributing to tumor cell trans-endothelial migration and metastasis.

### 5.8. Fucosylation in Metastasis

The function of fucosylation in metastasis is not limited to facilitating EMT and trans-endothelial migration of tumor cells across blood vessels. The amount of circulating tumor cells (CTCs) shed from primary tumors as well as their rolling velocities along blood vessels are also dependent on fucosylation levels. For instance, in a simulated model, knockdown of *FUT3* significantly reduced the number of flowing cells under shear stress. High rolling velocity that disfavors settlement of flowing cells was also observed in cells with *FUT3* knockdown [141]. Moreover, knockdown of *FUT3* lowered expression of the lectin ligand SLe^x^, hence weakening interactions between CTCs and GBPs present on other cells [141,142]. As proper binding of CTCs toward GBPs is required for the attachment of tumor cells to secondary sites [142], the attenuated binding affinity to GBPs caused by *FUT3* knockdown is one example of the metastatic regulation of CTCs by fucosylation alteration.

In addition, fucosylation affects information transmission between tumor cells and other cells within TME. This behavior, primarily regulated by tumor-derived exosomes, also facilitates metastasis. Exosomes are integrin-coated extracellular vesicles carrying miRNAs, mRNAs, long noncoding RNAs (lncRNAs), DNAs and lipids, and are primarily involved intercellular communication [143]. Although the mechanisms behind regulation of exosomes by fucosylation remain underexplored, evidence suggests an association between these two biological features. Compared to those from normal cells, exosomes derived from tumors display analogous features to their parent cells such as aberrant fucosylation. For instance, evidence has shown that integrins were over-expressed and heavily fucosylated in tumor-derived exosomes [144]. Meanwhile, others have demonstrated that fucosylated exosomes promote cancer growth while soluble fucosylated glycans had no effect on cell proliferation [145]. Moreover, exosomes derived from tumor cells could be used to relieve cellular stress to maintain cell growth and survival. Evidence suggests that tumor cells sequester miRNAs that disfavor tumor growth into fucosylated exosomes to mitigate the suppressive effects of these miRNAs [146].

Taken together, fucosylation status significantly affects metastasis in almost every step of the process including EMT, detachment of tumor cells from primary tumor, intravasation, CTCs rolling along blood vessels, extravasation, and settlement to secondary sites. The most well-characterized mechanism of fucosylation contribution to metastasis is through intercellular interactions between fucosylated surface molecules and tumor cells or tumor-derived exosomes with GBPs presented on the surface of other cells. Despite this, studies have been limited to the effects of FUTs on fucosylation-dependent metastasis promotion, while the function and relevance of fucosidases remain underexplored.

### 5.9. Fucosylation in Immune Surveillance

Fucosylation also participates in the escape of tumor cells from immune surveillance. The innate immune response involves binding of receptors on immune cells to antigens presented on target cells. The binding and function of either the immune receptors or target cell antigens can be modulated by glycosylation, including fucosylation. Often tumor cells utilize glycosylation to mask their foreignness and escape from immune surveillance [39]. For example, fucosylation of tumor cell antigens promotes their recognition by Natural Killer (NK) cells [147]. Accordingly, some cancer cells have reduced GDP-fucose production due to a mutation in *GMDS* [148,149], leading to a reduction in the abundance of Lewis antigen. This attenuates NK cell responses that depend on recognition of antigen, including activation of tumor necrosis factor (TNF) receptor superfamily member 6 (CD95) and TNF-related apoptosis-inducing ligand (TRAIL)-induced apoptosis pathways, thus protecting tumor cells from immune cell attack [148,150].

These results demonstrate that fucosylation is essential for antigen presentation on the cell surface, and when compromised can enable tumor cells to evade the immune response.

### 5.10. Fucosylation in Multidrug Resistance (MDR)

MDR is a major obstacle to cancer treatment, as it results in resistance to chemotherapy [39]. Fucosylation has been linked to the acquisition of MDR in tumor cells, suggesting that it could be targeted as an adjunct to chemotherapy. Cells that have acquired MDR often display N-linked glycans with high levels of core fucosylation [151], attributed to altered FUT8 and FUCA1 activities [7,126,137,152,153]. Furthermore, enhanced expression of FUTs observed in cancers can contribute to MDR development by activating the PI3K/Akt and extracellular signal-regulated kinase (ERK)/MAPK pathways, which facilitate the survival of cells and compromise the efficacy of anticancer therapies [154,155]. In addition, MDR-associated protein 1 (MRP1), which promotes MDR, has been reported to be regulated by FUTs [154].

In summary, fucosylation can modulate the functional properties of proteins and has been linked to carcinogenesis and cancer progression, similar to other glycosylation processes. One major effect is the regulation of Lewis antigen synthesis, affecting their interaction with lectin receptors on immune cells to escape from immune surveillance and on endothelial cells to promote tumor migration [147,148,149]. The strong correlation between fucosylation levels and cancer progression suggests that these fucosylation-related enzymes could be potential therapeutic targets for new cancer treatments. This may include inhibition of FUTs to decrease fucosylation [156], or enhancement of FUCA1 activity, which has been speculated to possess tumor-suppressive potential.

## 6. Regulation of Fucosylation by the p53-FUCA1 Axis

Investigations of the tumor-suppressive potential of FUCA1 led to the discovery that its gene is a transcriptional target of p53. Thus, p53, in addition to its other reported functions, appears to promote cancer by regulating fucosylation [7].

### 6.1. Transcriptional Regulation of FUCA1 by p53

The idea that *FUCA1* is regulated by p53 was first proposed based on the observed co-expression of p53 protein and *FUCA1* mRNA [7,157]. This was confirmed by the observation that p53 protein binds to the *FUCA1* gene [7] and that proper binding of p53 to its responsive elements on *FUCA1* is necessary for the transcription activation of *FUCA1* [157]. Thus, *FUCA1* was deemed a direct transcriptional target of p53. Moreover, chemotherapies that induce p53-dependent pathways upregulate *FUCA1* mRNA expression in a p53-dependent manner [157]. However, *FUCA1* does not respond to regulation by p73, a p53 family member that shares many transcription targets with p53 [157]. These results indicate that transcriptional control of *FUCA1* by p53 is innate and specific to endogenous p53.

Confirmation of the transcription regulation of *FUCA1* by p53 is not limited to in vitro but also extends to in vivo studies. In specimens collected from breast cancer patients bearing wild-type p53, *FUCA1* expression is high, whereas specimens containing mutant p53 show low levels of *FUCA1* [7]. The same trend was observed in thyroid cancer patients, as follows: *FUCA1* RNA expression is high in papillary thyroid cancers (PTCs) which mostly carry wild-type p53, while low levels of *FUCA1* RNA were detected in anaplastic thyroid cancers (ATCs) where p53 is frequently mutated. These observations indicate that regulation of *FUCA1* transcription is dependent on p53 status (wild-type or mutated). In highly aggressive cancer types, common phenotypes include the loss of wild-type p53, which could explain the observed correlation between cancer aggressiveness and the difference in *FUCA1* expression in ATCs that are derived from more differentiated PTCs [158].

### 6.2. Regulation of Fucosylation by the p53-FUCA1 Axis

FUCA1 fucosidase activity is also positively corelated with the extent of p53 induction [157]. FUCA1 fucosidase activity can be upregulated by DNA damage and overall fucosylation levels are decreased in cells having wild-type p53 [7]. Moreover, fucosidase activity can be induced by exogenous p53 overexpression, and this is further potentiated by the additional induction of DNA damage [7]. Similarly, increases in the levels of FUCA1 proteins was observed following p53 induction, whereas knocking down *FUCA1* led to a loss of overall fucosidase activity in the cell [157]. Taken together, these results indicate that FUCA1 fucosidase activity as well as cellular fucosylation levels are both regulated by p53.

### 6.3. Tumor Suppression by the p53-FUCA1 Axis

This relationship with p53 suggests that FUCA1 could function to suppress tumors. This is supported by the observation that p53 exhibits a greater impact on tumor suppression compared to p73 [157], consistent with *FUCA1* being a specific target of p53 but not p73.

The p53-FUCA1 axis have been shown to play a role in the induction of apoptosis [7]. Overexpression of FUCA1 inhibits cell proliferation by increasing apoptosis in breast cancer, CRC, GBM, and lung cancer cells, independent of p53 status [7]. FUCA1 was shown to reduce fucosylation of EGFR by catalyzing α1,6-defucosylation (Figure 6). The resulting decrease in EGFR fucosylation in turn represses phosphorylation of both EGFR and Akt [7] (Figure 6), a downstream effector of the EGFR pathway, indicating attenuated activity of both EGFR and Akt. The observation that expression of FUCA1 inhibits the EGFR pathway, implicated in promoting cancer, again supports the potential role of FUCA1 as a tumor suppressor.

Another study has shown that FUCA1 fucosidase plays a role in p53-mediated apoptosis in an osteosarcoma cell line previously subjected to chemotherapy [157]. This study showed that while overexpression of FUCA1 does not affect cancer cell viability, knockdown of *FUCA1* impairs the ability of p53 to induce activation of effector caspases that would otherwise cause apoptosis and cell death [157]. On the other hand, other studies have demonstrated that overexpression of FUCA1 does not affect the clonogenic potential of thyroid cancer or normal cells [158].

These results demonstrate the tumor suppressing potential of FUCA1 is at least partially dependent on its regulation by p53. On the other hand, the inconsistent effects of FUCA1 on cancer cell viability show that the mechanisms linking FUCA1 and cancer progression are as yet not fully understood. As such, further investigation on FUCA1, fucosylation and p53 would improve our understanding of cancer development from the aspect of protein glycosylation and could provide novel targets for cancer therapies.

## 7. Conclusions and Future Directions

Glycoconjugates are present in every cellular compartment and can play a critical role in determining the function of biomolecules. The enzymatic regulation of glycosylation and the generation of multiple glycoforms play an important role in many cellular mechanisms, including the development of cancer [1]. Glycosylation patterns often change when cells are subjected to stress [2]. Understanding these changes and their mechanistic consequences may explain the relationship between glycosylation and cancer development. These mechanisms include receptor–ligand binding, cell–cell adhesion, extracellular interaction and communication within the tumor microenvironment. There are a number of unexplored areas here. For instance, although a correlation between the expression of the p53 target *FUCA1* and cancer has been reported, the precise mechanism that underlies this correlation is not well understood [7,116,125,126]. Understanding the mechanisms by which altered glycosylation can contribute to tumorigenesis, cell migration, metastasis, immune escape and the drug resistance of cancer cells may inform the development of cancer therapeutics that modulate glycosylation. In addition, investigation into cancer-promoting glycan structures and the responsible enzymes could enhance our understanding of the relationship between cancer development and glycosylation [1,2,24,37,39,52,60,159]. Moreover, utilizing bioinformatic strategies such as metabolomics, glycomics and glycoproteomics could unravel more biomarkers for cancer diagnosis and prognosis, as well as identify novel candidates for cancer drug development.

## Figures and Tables

**Figure 1 cancers-16-02753-f001:**
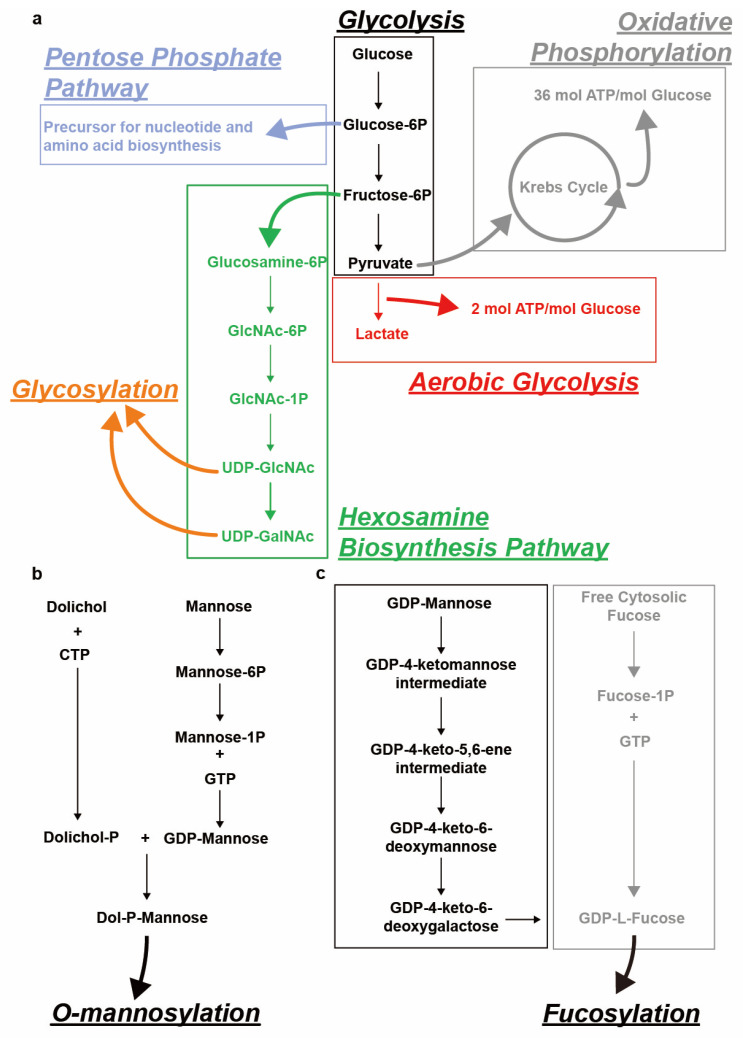
Schematical representation of sugar nucleotide biosynthesis for various types of glycosylation. (**a**) Uridine diphosphate N-acetylgalactosamine (UDP-GalNAc), UDP-N-acetylglucosamine (UDP-GlcNAc) biosynthesis from glucose via the glycolysis-related hexosamine biosynthesis pathway (HBP). Other glycolysis-related pathways including the pentose phosphate pathway, aerobic glycolysis and oxidative phosphorylation are also depicted. (**b**) Dolichol-phosphate-mannose (Dol-P-Man) biosynthesis from Dolichol and mannose. (**c**) Guanosine diphosphate fucose (GDP-fucose) biosynthesis from guanosine diphosphate mannose (GDP-mannose) (de novo synthesis) and free cytosolic fucose (salvage pathway).

**Figure 2 cancers-16-02753-f002:**
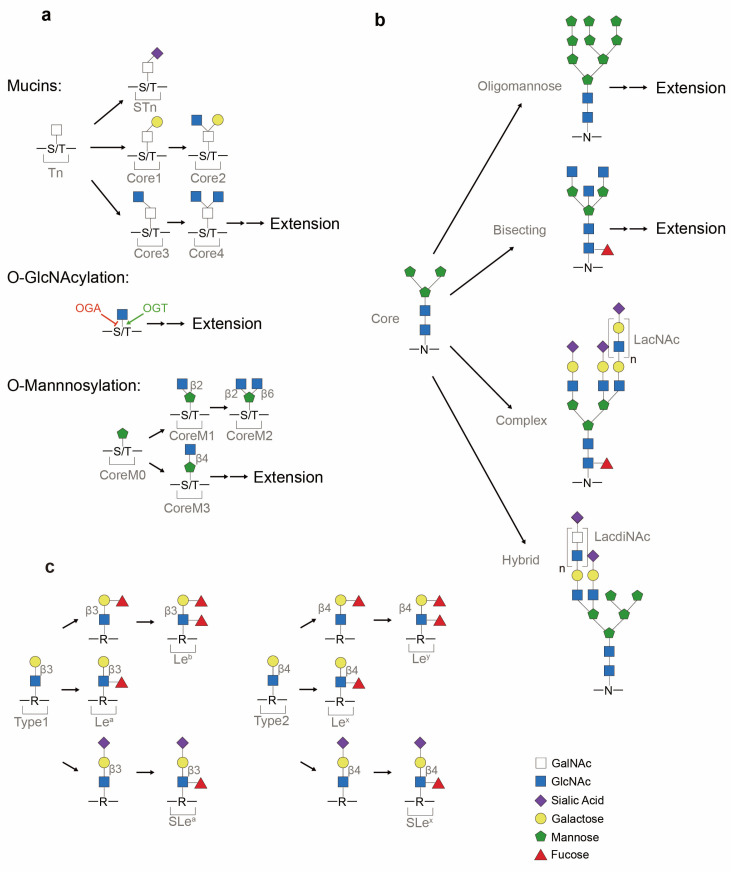
Representative glycans common in protein glycosylation. (**a**) O-linked (mucins, O-GlcNAcylated and O-mannosylated) glycosylation. Core structures of O-GalNAcylated mucins consist of Thomsen-nouveau (Tn) antigen, Thomsen–Freidenreich antigen (TF antigen or T antigen or core 1), core 2, core 3, core 4 and sialyl-Tn (STn). After adding the first GalNAc to serine/threonine (S/T), this monosaccharide-linked structure is termed the Tn antigen. If galactose is added to the Tn antigen, it will result in the formation of core 1. If a sialic acid is attached to Tn, a terminal structure STn is formed. O-GlcNAcylation of proteins is balanced between O-GlcNAc transferase (OGT) and O-GlcNAcase (OGA). O-mannosylation contains the major structures of coreM0, coreM1, coreM2 and coreM3. Linkage between GlcNAc and mannose differs CoreM1 (β1-2 attachment) and CoreM3 (β1-4 attachment) structures. (**b**) Asparagine (N)-linked (N-GlcNAcylated) glycosylation. Mature N-glycan precursor is formed from the Man_3_GlcNAc_2_ core into oligomannose, complex and hybrid glycans. Extension by the Galβ1-4GlcNAc building block (LacNAc sequence) or GalNAcβ1-4GlcNAc building block (LacdiNAc sequence) is commonly observed in glycans. (**c**) Terminal glycan structures (Lewis antigens). Common precursors of Type 1 and Type 2 chains are synthesized by a β1,3-GlcNActransferase which catalyzes the attachment of β-GlcNAc block onto glycans and glycolipids (represented by R). Attachment of the outermost galactose defers Type 1 (β1-3 attachment) and Type 2 (β1-4 attachment) glycans. From Type 1 and Type 2 backbones, Lewis antigens, assisted by fucosyltransferases (FUTs) and sialyltransferases, are differentiated into Le^b^, Le^a^, SLe^a^, Le^y^, Le^x^ and SLe^x^.

**Figure 3 cancers-16-02753-f003:**
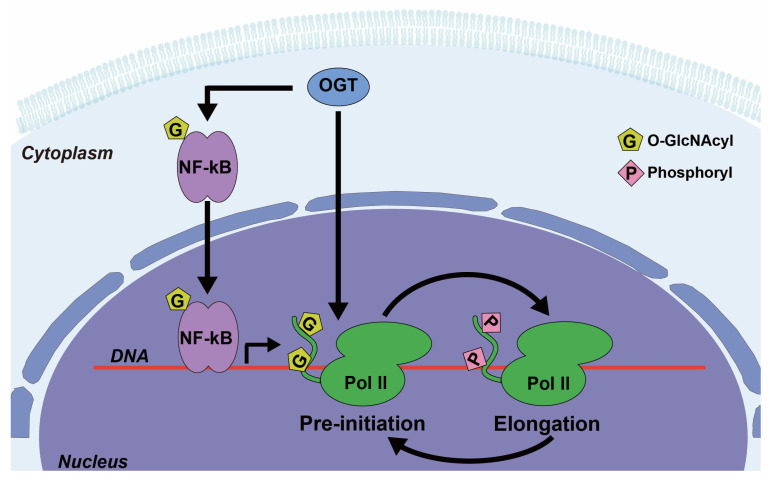
Examples of O-GlcNAc transferase (OGT) involvement in the regulation of various signaling pathways. OGT catalyzes the addition of O-GlcNAc to nuclear factor kappa B (NF-κB), which enhances its nuclear localization and transcriptional activity. OGT also adds O-GlcNAc to the C-terminal domain (CTD) of RNA polymerase II (Pol II), which competes with phosphorylation at the same sites and can switch Pol II between the pre-initiation and the elongation states.

**Figure 4 cancers-16-02753-f004:**
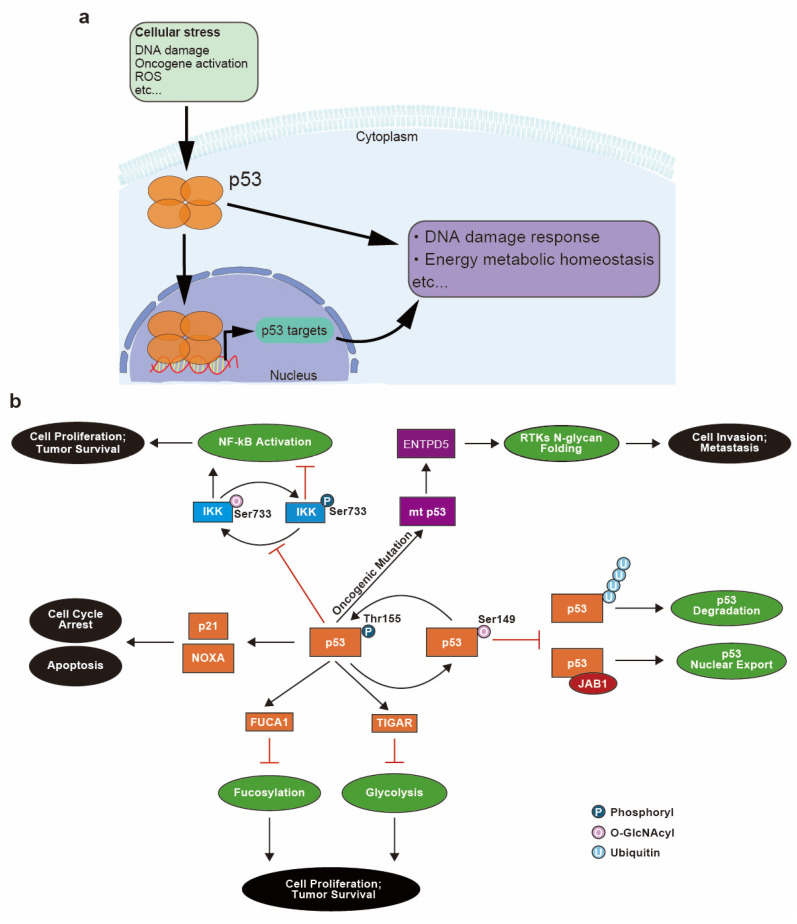
Involvement of p53 in cell stress responses and glycosylation-related pathways and the relationship between p53 glycosylation and cancer progression. (**a**) Upon cellular stress, p53 is upregulated and translocated into the nucleus to transcriptionally regulate genes that are implicated in stress response pathways including the DNA damage response and energy metabolism homeostasis. (**b**) During the response to DNA damage, wild-type p53 is phosphorylated and activated, which leads to cell cycle arrest and apoptosis via transcription of *p21* and *NOXA*. p53 also transcriptionally regulates expression of alpha-L-Fucosidase 1 (*FUCA1*) and *TP53*-induced glycolysis and apoptosis regulator (TIGAR) protein to modulate fucosylation and glycolysis pathways. Hyperactivation of these two pathways is associated with cell proliferation and tumor survival. p53 inhibits pro-proliferative NF-κB activation by favoring phosphorylation (Ser733) of Inhibitor of nuclear factor-κB (IκB) kinase (IKK), which competes with O-GlcNAcylation (Ser733) of IKK. O-GlcNAcylation of p53 (Ser149) competes with phosphorylation (Thr155), which attenuates ubiquitination-mediated p53 degradation as well as p53 nuclear export mediated by Jun activation domain-binding protein 1 (JAB1). Oncogenic mutant of p53 (mt p53) also promotes cancer by inducing N-glycan folding of receptor tyrosine kinases (RTKs) via the transcriptional regulation of ectonucleoside triphosphate diphosphohydrolase 5 (*ENTPD5*) protein.

**Figure 5 cancers-16-02753-f005:**
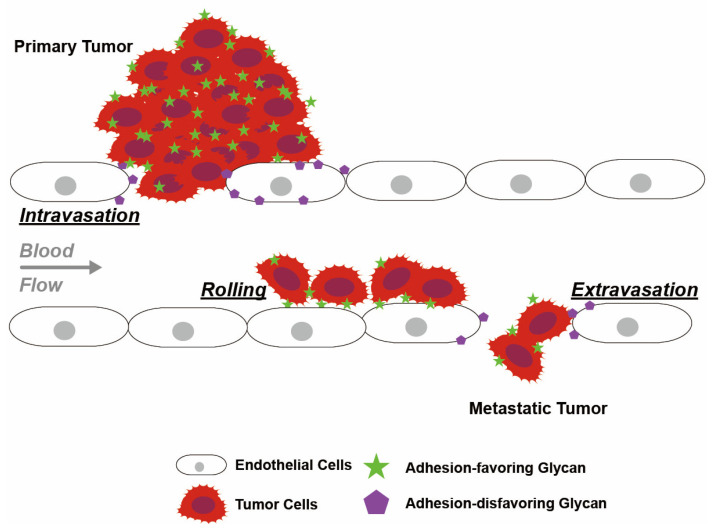
Glycan-mediated intravasation, rolling and extravasation of tumor cells along the endothelium. Adhesion-disfavoring glycans attenuate adhesion and reduce cell–cell interactions with the endothelium and enhance endothelial permeability, thereby facilitating tumor intravasation and extravasation. Glycans on tumor cells bind to lectin receptors on endothelial cells with low affinity, which contributes to tumor cell rolling along the endothelium. Glycan structures involved in this process include Lewis antigens and lectin receptors.

**Figure 6 cancers-16-02753-f006:**
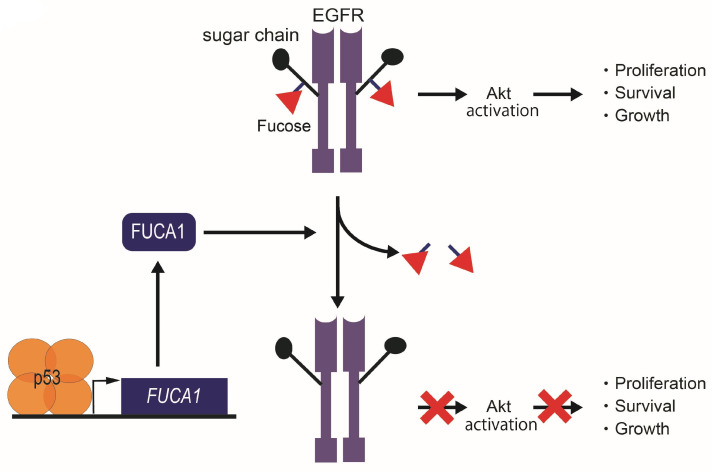
Tumor suppression function in cancer cells by the p53-FUCA1 axis. FUCA1 is one of the p53 target genes. FUCA1 reduces α1,6-fucosylation of EGFR and inhibits the Akt signaling pathway in cancer cells.

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
