# Peer review of "FUCA1: An Underexplored p53 Target Gene Linking Glycosylation and Cancer Progression"

_cancers, 2024, doi:10.3390/cancers16152753_

Round 1

Reviewer 1 Report

Comments and Suggestions for Authors

This review summarizes the role and mechanism of glycosylation in cancer, as well as the role of tumor suppressor p53 in the regulation of glycosylation as a potential tumor-suppressive mechanism.  Further understanding the mechanistic relationships between glycosylation and cancer progression may provide important insights into the development of novel and effective cancer treatments. Overall this manuscript is very well written and logically presented, which will be of interest to the broad audience in the metabolism and cancer research field.

Very minor points:

There are some typos. For instance:

1.       Figure 6. Tumor suppression function in cancer cell (cells) by the p53-FUCA1 axis. FUCA1 is one of the p53 target gene (genes). FUCA1 reduced (reduces) α1,6-fucosylation of EGFR and inhibits Akt signaling pathway in cancer cells.

2.       Table 1. Summary of major glycosylation-related enzymes together with their enzymatic function (functions), implication in glycosylation process and cancer indication.

3.       Table 2. Summary of glycosylation alterations that is (are) induced during tumor microenvironment establishment and maintenance.

Comments on the Quality of English Language

The quality is good. A few typos as shown above

Author Response

First of all, we thank the reviewer very much for his/her careful consideration of our manuscript, and we appreciate the reviewer’s helpful comments. The responses are outlined below (in blue type).

Very minor points:

There are some typos. For instance:

  1. Figure 6. Tumor suppression function in cancer cell (cells) by the p53-FUCA1 axis. FUCA1 is one of the p53 target gene (genes). FUCA1 reduced (reduces) α1,6-fucosylation of EGFR and inhibits Akt signaling pathway in cancer cells.

  1. Table 1. Summary of major glycosylation-related enzymes together with their enzymatic function (functions), implication in glycosylation process and cancer indication.

  1. Table 2. Summary of glycosylation alterations that is (are) induced during tumor microenvironment establishment and maintenance.

We thank the reviewer’s comments and have corrected the typos accordingly (shown in yellow).

Reviewer 2 Report

Comments and Suggestions for Authors

This review by Hu et al., discusses about the glycosylation links with the cancer development and regulatory role of p53-dependent fucosylation factor FUCA1. Authors first introduced about basics of glycosylation and metabolism with schematics, tables, and then establish their links with signaling pathways and oncogenesis. Then discuss about p53 pathways, glycosylation links and finally covers fucosylation-p53 axis and its importance in cancer.

Major:

This review provides a comprehensive and detailed summary of glysosylation and the fucosylation-p53 axis, but I couldn't find enough information concerning the FUCA1-p53 axis and its relationship to glycosylation regulation in cancers. Correlation of FUCA1-p53 axis with cancer progression is convincing and supported with appropriated references.

 Minor:

Proofing needed, for eg ref 150 and 151 repeated twice. 

Comments on the Quality of English Language

no issues with the english language quality.

Author Response

We sincerely appreciate the comments by the reviewer.  The responses are outlined below (in blue type).

Major: 

This review provides a comprehensive and detailed summary of glysosylation and the fucosylation-p53 axis, but I couldn't find enough information concerning the FUCA1-p53 axis and its relationship to glycosylation regulation in cancers. Correlation of FUCA1-p53 axis with cancer progression is convincing and supported with appropriated references.

We understand the reviewer’s concern about the relative lack of information regarding the FUCA1-p53 axis and its relationship to glycosylation. However, it must be noted that FUCA1 is a new and relatively under-explored research subject, and all available information currently in the literature has been presented in the section titled “Regulation of Fucosylation by the p53-FUCA1 Axis.”

As the reviewer pointed out, the importance of fucosylation in cancer progression is well recognized. Current studies mostly focus on FUTs that catalyze the addition of fucose. Meanwhile, the FUCA1-p53 axis is a lesser-known pathway that could modulate the cellular fucosylation profile in parallel to FUTs. We aim to spark interest in the FUCA1-p53 axis in this review to encourage future research on this pathway.

Minor: 

Proofing needed, for eg ref 150 and 151 repeated twice. 

Ref 151 is corrigendum of Ref 150, as the same case for Ref 23 and Ref 24. We have deleted ref 151 and 23. To make sure of not repeating the references, we checked and corrected the reference list again (shown in yellow).

Reviewer 3 Report

Comments and Suggestions for Authors

Cancer is a challenging disease to cure, largely due to drug resistance and disease relapse. Glycosylation, a modification on cellular biomolecules, has been linked to carcinogenesis by influencing cellular functions. Recent discoveries have highlighted the role of p53 in energy metabolism and its target gene, alpha-L-fucosidase 1 (FUCA1), which is essential for fucosylation. Glycan structures and glycosylation-related enzymes play crucial roles in cancer development, impacting cell-matrix interactions, cell-cell adhesion, and cell migration. The review discusses the interplay between glycosylation and tumor microenvironmental factors, along with its involvement in cancer-promoting mechanisms like EGFR, PI3K/Akt, and p53 pathways. Further investigation into glycosylation's mechanistic relationships may lead to novel cancer treatments, although an exploration of FUCA1's role in regulating circulating cells and metastasis would be beneficial.

1. Authors should discuss the role of FUCA1 in regards to exosomes regulation. 

2. Authors should also discuss the role of FUCA1 in circulating cell and in development.  

3. Do stem cells regulate FUCA1 ? 

Author Response

Reviewer #3:

We sincerely appreciate the constructive criticisms raised by the reviewer and we believe that our manuscript has been significantly improved as a result.  The responses are outlined below (in blue type).

  1. Authors should discuss the role of FUCA1 in regards to exosomes regulation. 
  2. Authors should also discuss the role of FUCA1 in circulating cell and in development.  
  3. Do stem cells regulate FUCA1? 

Based on the reviewer’s suggestions, we discussed the role of fucosylation in circulating cells and exosome regulation is discussed in a new section titled “Fucosylation in Metastasis” from Line 434-466 (shown in yellow).

In addition, discussion regarding the effects of fucosylation on embryo development as well as cancer and normal stem cells have been discussed in a new section titled “Fucosylation in Cancer Stem Cells” from Line 373-392 (shown in yellow).

As the exact roles of FUCA1 and fucosidase in these aforementioned processes have not been fully elucidated, we also presented information on the general fucosylation to supplement the topic.

Round 2

Reviewer 2 Report

Comments and Suggestions for Authors

The authors have clarified and addressed my comments in the revised manuscript. I do not have any further suggestions. Best wishes!